# Rapid Point-of-Care PCR Testing of Drug-Resistant Strains on Endotracheal Aspirate Samples: A Repurposed Effective Tool in the Stepwise Approach of Healthcare-Acquired Pneumonia—A Pilot Study

**DOI:** 10.3390/ijms241713393

**Published:** 2023-08-29

**Authors:** Andrei-Mihai Bălan, Constantin Bodolea, Andrada Nemes, Rareș Crăciun, Natalia Hagău

**Affiliations:** 1Department of Anaesthesia and Intensive Care 2, “Iuliu Hațieganu” University of Medicine and Pharmacy Cluj-Napoca, 400012 Cluj-Napoca, Romania; balan.andrei@umfcluj.ro (A.-M.B.); cbodolea@gmail.com (C.B.); hagaunatalia@gmail.com (N.H.); 2Department of Anaesthesia and Intensive Care, Municipal Clinical Hospital, 400139 Cluj-Napoca, Romania; 3Department of Internal Medicine, “Iuliu Hațieganu” University of Medicine and Pharmacy Cluj-Napoca, 400012 Cluj-Napoca, Romania; craciun.rares.calin@elearn.umfcluj.ro; 4Gastroenterology Clinic, ”Prof. Dr. O. Fodor” Regional Institute of Gastroenterology and Hepatology, 400162 Cluj-Napoca, Romania; 5Department of Anaesthesia and Intensive Care, ”Regina Maria” Hospital, 400221 Cluj-Napoca, Romania

**Keywords:** pneumonia, polymerase chain reaction, rapid diagnosis, ventilator-associated pneumonia, intensive care-acquired pneumonia, multidrug resistance

## Abstract

Healthcare-associated pneumonia (HCAP) is a common nosocomial infection with high morbidity and mortality. Culture-based detection of the etiologic agent and drug susceptibility is time-consuming, potentially leading to the inadequate use of broad-spectrum empirical antibiotic regimens. The aim was to evaluate the diagnostic capabilities of rapid point-of-care multiplex polymerase chain reaction (PCR) assays from the endotracheal aspirate of critically ill patients with HCAP. A consecutive series of 29 intensive care unit (ICU) patients with HCAP and a control group of 28 patients undergoing elective surgical procedures were enrolled in the study. The results of the PCR assays were compared to the culture-based gold standard. The overall accuracy of the PCR assays was 95.12%, with a sensitivity of 92.31% and a specificity of 97.67%. The median time was 90 min for the rapid PCR tests (*p* < 0.001), while for the first preliminary results of the cultures, it was 48 h (46–72). The overall accuracy for rapid PCR testing in suggesting an adequate antibiotic adjustment was 82.98% (95% CI 69.19–92.35%), with a specificity of 90% (95% CI 55.50–99.75%), a positive predictive value of 96.77% (95% CI 83.30–99.92%), and a negative predictive value of 56.25 (95% CII 29.88–80.25%). This method of rapid point-of-care PCR could effectively guide antimicrobial stewardship in patients with healthcare-acquired pneumonia.

## 1. Introduction

Pneumonia is a clinical condition characterized by the infection of the inferior respiratory tract localized in the pulmonary parenchyma, and it is associated with relatively high mortality and a significant healthcare-associated burden [1,2]. The most clinically relevant distinction is between community-acquired pneumonia (CAP) and healthcare-associated pneumonia (HCAP), which provides an initial support tool for empirical therapeutic management. Classically, HCAP comprises three different entities:Pneumonia that develops in nonhospitalized patients in nursing homes and extended care facilities or in patients undergoing home infusion therapies, chronic dialysis, and other similar scenarios.Hospital-acquired pneumonia (HAP)—onset during hospitalization, after a minimum of 48 h from admission, on the regular ward.Ventilator-associated pneumonia (VAP)—arising in patients requiring invasive mechanical ventilation for a minimum of 48 h [1,2,3,4,5,6].

ICU-acquired pneumonia (ICU-AP) is a relatively new concept, defined as pneumonia affecting patients admitted to the ICU for at least 48 h. Furthermore, this can be classified as ventilator-associated pneumonia and hospital-acquired pneumonia in ICU patients, which can be nonventilated (NV-ICU-HAP) and ventilated hospital-acquired pneumonia (V-ICU-HAP) [7,8].

HAP is the second-most common nosocomial infection after urinary tract infections. Some studies estimate 5–20 cases of pneumonia for every 1000 admissions, and it is a leading cause of death among patients who ultimately require ICU admission [2]. A recent study estimated the incidence of NV-ICU-HAP to be 4.5 per 1000 patient ICU days compared to V-ICU-HAP 21 per 1000 invasive mechanical ventilation days [8,9]. All-cause mortality, especially with VAP, ranges among studies between 20% and 50%. While the directly attributed mortality might be a matter of debate, it appears to be up to 13%, according to some studies [2,4,7,8,10,11,12]. Respiratory tract infections, like HAP/VAP or ICU-AP, can be polymicrobial or caused by multidrug-resistant (MDR) bacteria; therefore, initiation of an early and as-targeted-as-possible antibiotherapy is essential to avoid the risk of complications and death [4,13,14,15,16]. Aliberti et al. showed a high prevalence of MDR bacteria among patients with pneumonia admitted to ICUs, especially if they receive mechanical ventilation [17].

The most-used microbiological diagnostic technique (the classic plate cultures) usually requires a minimum of 24–48 h for bacterial identification while emitting a correct antibiogram. Meanwhile, the patient receives broad-spectrum antibiotics, usually a combination approach that can be dangerous for the patient because it can lead to the development of MDR/extensively drug-resistant (XDR) or even pan-drug-resistant (PDR) bacteria, notwithstanding the side effects and complications associated with the bacteria itself (*Clostridioides difficile* colitis, for example) [2,3,18,19].

A fast method for pathogen identification is based on molecular techniques, like PCR [1,3,20]. It has been shown that for pneumonia diagnosis, when using classic culture-based methods, the causative pathogen is found in only 5–10% of the cases, mainly because of the previous administration of antibiotics. In the case of molecular-based diagnosis, like PCR techniques, correct identification of the causative agent can reach 50–75% of cases [2,21]. The development and employment of microfluidic systems allowed PCR devices to be small, compact, and rapid, making them adequate for usage at the patient’s bedside—point-of-care (POC) [22,23].

Cepheid’s GeneXpert device (Cepheid, Sunnyvale, CA, USA) is a point-of-care system with all the essential steps automated and integrated, making it very easy to use. This device offers the results in approximately 1 h for most tests and costs about 40 EUR per sample [24,25]. One of the main knowledge gaps for this device regarding its use in managing patients with pneumonia is the lack of studies, especially with samples taken directly from the lower respiratory tract.

The current study aimed to assess the diagnostic performance of a rapid POC PCR device (Cepheid GeneXpert, assays: Xpert^®^ Carba-R, Xpert^®^ MRSA/SA SSTI, Xpert^®^ vanA/vanB, Cepheid, Sunnyvale, CA, USA) in detecting drug-resistant strains from the endotracheal aspirate (ETA) of critically ill patients with HCAP, compared to the culture-based gold standard of care. As secondary objectives, the study aimed to assess whether rapid PCR results could improve antimicrobial stewardship in this clinical scenario.

## 2. Results

### 2.1. Baseline Characteristics

A total of 57 patients were included in the study, and 81 GeneXpert tests were performed. Of the 57 patients, *n* = 28 (49.1%) were in the elective surgery control group, and *n* = 29 (50.9%) were in the HAP/VAP group. The baseline characteristics of the patients included in the HAV/VAP group are reported in Table 1, along with the essential characteristics of the control group (if applicable). The in-hospital mortality for the pneumonia group was 42.1% (n = 16/29).

### 2.2. Infection Profile

All patients in the control group had negative cultures, while all patients in the HAP/VAP group had positive cultures with at least one multidrug-resistant bacteria. Five patients (17.24%) had a second concomitant positive isolate of multidrug-resistant bacteria. The bacteria isolates and sensibility profiles of the culture isolates are summarized in Table 2. Of the 29 patients with HAP/VAP, the clinical suspicion was based on positive chest imaging (79.3%). The patients with negative imaging had progressive worsening of the ventilation parameters, purulent respiratory secretion, hemodynamic instability, or worsening of the inflammation profile.

### 2.3. Diagnostic Capabilities of the Sample-to-Answer PCR Test

There was a total of 81 PCR tests performed, of which 28 (34.56%) were in the control group. In the study group, the selection of the specific mutation assay was chosen based on rapid Gram staining, and the results were subsequently compared to the cultures and drug-resistance profiles as a gold standard. The median time to the first preliminary results of the cultures was 48 h (46–72), compared to a median of 90 min for the rapid PCR tests (*p* < 0.001). The mean cycle threshold level was 28.95 ± 6.42, with a mean end point of 320.26 ± 158.06.

All the cultures and PCR assays in the control group were negative. The overall accuracy of the PCR assays was 95.12%, with a sensitivity of 92.31% and a specificity of 97.67%. The different assay kits were also assessed individually. The overall diagnostic prowess of the sample-to-answer PCR test is depicted in Table 3.

### 2.4. The Potential Therapeutic Impact of the Rapid PCR Tests

All the patients with suspected pneumonia were started on an empiric antibiotic regimen based on the hospital protocols, and the regimen was adjusted following the results of the cultures and corresponding antibiograms. The antibiograms imposed the adjustment of the antibiotic regimen in n = 19 (65.51%) patients. There were 39 distinct bacteria isolated in the pneumonia group, of which antibiotic adjustments according to the antibiogram were imposed 29 times (74.33%). Of the 29 adjustments, n = 22 (75.86%) were accurately predicted by a rapid PCR test, with a median of 46 (44–72) hours before the culture results. The overall accuracy for rapid PCR testing in suggesting an adequate antibiotic adjustment was 82.98% (95% C.I. 69.19%–92.35%), with a specificity of 90% (95% C.I. 55.50–99.75%), a positive predictive value of 96.77% (95% C.I. 83.30–99.92%), and a negative predictive value of 56.25 (95% C.I.I 29.88–80.25%). Within the 28 instances of positive results using the carbapenem-resistance kit, there were n = 14 (50%) KPC mutations, n = 7 (25%) NDM mutations, and n = 7 (25%) OXA-48 mutations. The PCR test correctly identified 7 instances of carbapenem resistance in patients on meropenem or imipenem; 11 instances of beta-lactam resistance in patients on piperacillin–tazobactam (n = 5), cephalosporins (n = 5), or ampicillin/sulbactam (n = 1); and 3 instances of vancomycin resistance in patients on vancomycin.

## 3. Discussion

To our knowledge, this is the first repurposing attempt for rapid point-of-care PCR assays using ETA samples in mechanically ventilated patients with HAP/VAP, providing an essential proof-of-concept for a method designed for other clinical scenarios. The results of the current pilot study suggest that the method is an effective tool in facilitating antimicrobial stewardship in HAP and VAP, especially in a setting with a high prevalence of MDR and XDR strains. Compared to the gold standard of ETA cultures, the PCR assays had a 95.12% accuracy, with a sensitivity of 92.31% and a specificity of 97.67% in detecting specific drug-resistant strains, providing the results quicker within a median of 46 h, compared to ETA cultures.

While there are significant differences in the design of our study compared to studies previously published in the literature, our results are concordant with prior evidence. Regardless of the site, an MDR or XDR bacterial infection in the critically ill is associated with a very high mortality rate. All the HAV/VAP cases in our study were determined at least by MDR bacteria, and the patients had an in-hospital mortality of 42.1%. While the exact figures differ based on study protocol and clinical scenario, the high mortality rate is similar to other reports. A meta-analysis that included 2462 patients with CRE *Klebsiella pneumoniae*, of which almost 50% were admitted to the ICU, reported an overall mortality of 42.14% and a 54.30% mortality in patients with bloodstream infections [26]. Another small-scale retrospective study on KPC-producing *Klebsiella pneumoniae* VAP, which included 39 patients, reported a 30-day mortality of 33.3%, significantly higher compared to a control group comprising VAP with carbapenem-susceptible strains. An Italian study group reported a 77.8% 30-day mortality in 115 non-COVID-19 MDR infections, with *Acinetobacter baumanii* colonization being an independent predictor for mortality [27]. Another study retrospectively analyzed 60 patients with *Acinetobacter baumanii* MDR, XDR, and pan-drug-resistant VAP and reported a 63.3% mortality. However, the drug-resistance profile did not significantly influence the mortality [28].

Moreover, the extreme intrinsic severity of MDR and XDR infections is further increased by a delayed diagnosis of drug resistance, conventionally dependent on the cultures and antibiograms, with the quickest available preliminary results being typically available at 48 h post-sampling. A multicenter study conducted on the east coast of America that included 121 ICU patients with CRE infections reported in 2017 a delay between bacteremia onset to active antibiotic therapy of 47 h in a cohort with an overall mortality of 49% [29]. A more recently published study from the same group reported a significantly shorter time to active antibiotic receipt by using rapid PCR testing to identify KPC-positive strains (median 24 vs. 50 h, *p* = 0.009), thus effectively decreasing 30-day mortality from 47% to 24% (*p* = 0.007) [30]. Our pilot study suggests a similar pattern, with a delay between the PCR results and cultures of 46 h median, with 74% of the patients potentially benefiting from a quicker switch to active antimicrobial therapy. However, probably the most significant caveat of our research (inherent to its design as a pilot study) is its lack of an interventional component, as according to our protocol, the antibiotic regimen was not altered based on the PCR testing. Therefore, the actual positive impact of a quicker antibiotic regimen adjustment can only be inferred based on other reports.

Multiple studies have shown an excellent diagnostic accuracy of multiplexed PCR assays in detecting antibiotic-resistant mutations, with some nuanced differences based on the device model, design, and intended use. A study using the BioFire^®^ FilmArray^®^ System (Salt Lake City, UT, USA) on 2207 samples reported an overall sensitivity above 96% for all mutations, with a 100% sensitivity and specificity for Van A/B and KPC mutations [31]. Another study using the Verigene^®^ Gram-Negative Blood Culture Nucleic Acid Test (Nanosphere Inc., Northbrook, IL, USA) reported an overall concordance between the rapid test and culture results of 96.3%, with an accuracy of 100% for *Acinetobacter baumanii* strains and 86.1% for *Klebsiella pneumoniae* [32]. The GeneXpert^®^ system (Cepheid^TM^, Sunnyvale, CA, USA) we used in our study has the background of being among the first assays used for quick MDR detection in tuberculosis, and there is extensive evidence attesting its diagnostic capabilities in that field [33,34,35]. Regarding its use in the ICU setting, there are three available kits for the identification of the most relevant drug-resistant mutations, namely the Carba-R, the MRSA/SA SSTI, and the vanA/vanB assay, all used in our pilot study. The efficacy of the Carba-R kit was tested on a sample of 150 enterobacterial isolates, with a sensitivity of 97.8% and a specificity of 95.3% [36]. The method was also tested in small-scale, real-life clinical scenarios, such as patients with abdominal sepsis admitted to the ICU [37] or in the emergency department [38] with slightly lower figures for sensitivity and specificity, yet still above 90%. A slight gap between experimental design and real-life data is expected, as this was also evident in our results, with a sensitivity for all kits at 92.31% and a specificity of 97.67%. To our knowledge, there is only one head-to-head comparison between the different PCR devices to this point. While the design per se was not constructed to provide a sample-by-sample comparison, there appear to be no significant differences among the different manufacturers [39].

One key aspect in deciding to adopt a new protocol is cost-effectiveness. Unfortunately, given the noninterventional design of our study, we could not construct an unbiased cost-effectiveness model. However, extrapolating previously published data from MDR tuberculosis using the GeneXpert system substantially reduces costs compared to the culture-based gold standard [40,41,42].

While the main strength of our study is that it is among the first reports to suggest the effectiveness of point-of-care rapid PCR assay in HAP/VAP in an ICU setting with an extremely high prevalence of MDR and XDR strains, our design has some significant limitations. First, the study was designed as a proof-of-concept protocol with a clinically driven aim to improve our care in the ICU. Therefore, the scale is small, limiting our results’ significance. Furthermore, we did not alter patient therapy based on the early PCR results in this initial pilot study, guiding the antibiotic therapy only on culture-based grounds. We considered an interventional approach unethical because it would have consisted in guiding therapy without prior evidence of its effectiveness. Consequently, given this approach, we could not evaluate whether a quicker diagnosis improves the outcomes or if it is a cost-effective strategy.

As future directions of research, one key element would be to assess the prognostic impact of a diagnostic strategy based on rapid POC PCR analysis vs. the culture-based gold standard. Constructing such a design in a randomized, controlled manner might pose substantial ethical dilemmas if mortality is used as the primary endpoint. Not least, a cost-effectiveness Markov analysis should provide the grounding for its clinical implementation. However, while not necessarily supporting its use in clinical practice, the positive results of our small-scale pilot study should support the implementation of larger-scale research protocols and even provide the required evidence for randomized head-to-head interventional designs, using patient outcome as the primary endpoint. Moreover, the rapidity and effectiveness of the method provide sufficient evidence to support proactive infection and drug-resistant surveillance in ICU settings if proven cost-effective, as POC PCR is significantly quicker than conventional techniques.

## 4. Materials and Methods

### 4.1. Study Design and Participants

The research protocol was designed as a pilot prospective, observational, and longitudinal repurposing study. A consecutive series of patients were enrolled between March 2017 and October 2018 from the ICU of a high-volume tertiary-care facility. The study was registered within the www.clinicaltrials.gov system, last update 22 June 2023, with the ID: NCT05928208.

As a pilot study, to adequately represent both positive and negative samples, we included two distinct groups of patients: a study group comprising patients with HAP or VAP and a control group of patients with no evidence of respiratory tract infection. The inclusion criteria for both groups are described below:

#### 4.1.1. The Study Group

The study group comprised mechanically ventilated ICU patients meeting the diagnostic criteria for HAP or VAP according to the most recent guidelines regarding the management of patients with healthcare-associated pneumonia [4,43], as follows:A minimum length of 48 h of hospital stay (HAP) or mechanical ventilation (VAP).Clinical suspicion of pneumonia based on the following:New or progressive lung consolidation on chest imaging;New onset of fever;Purulent respiratory secretions;New onset of leukocytosis or leukopenia;Worsening oxygenation;Surrogate criteria: hemodynamic status alterations, increase in other serum markers of systemic inflammation (C-reactive protein, procalcitonin, or presepsin).
Mechanical ventilation at the time of inclusion.

Patients with a prior history of pneumonia or other intrathoracic infections, either within the same hospitalization or the past 90 days, were excluded from the study population.

#### 4.1.2. The Control Group

The control group was designed to maximize the odds of clear-cut negative results. Therefore, the group comprised patients undergoing elective surgical procedures under general anesthesia and mechanical ventilation. The inclusion criteria were as follows:No clinical signs of respiratory tract or systemic infections at the time of admission (cough, shortness of breath, chest pain, sore throat, or fever);No prior history of chronic pulmonary disease (i.e., chronic bronchitis, chronic obstructive pulmonary disease, or bronchiectasis);No history of respiratory infections in the past four weeks;No history of antibiotic therapy in the past three months.

Furthermore, Gram staining from endotracheal swabs was performed before PCR testing, following the same protocol as patients in the study group (Figure 1). Patients with evidence of bacteria on the Gram slides were excluded from the control group.

### 4.2. Study Protocol

The core principle of the study was to compare the diagnostic accuracy of POC PCR testing compared to the culture-based gold standard for diagnosing HAP and VAP from the ETA samples of mechanically ventilated patients. The design structure of the study protocol is depicted in Figure 1.

#### 4.2.1. Endotracheal Aspirate and Gram Staining

ETA samples were collected from all the patients in the study using a standardized method. A sterile suction catheter was inserted through the endotracheal tube, and the aspirate was collected for analysis.

Following ETA sampling, Gram stain slides were prepared. Prior to plating, endotracheal swabs were evaluated qualitatively (with Gram stain coloration) using the Bartlett score, which implies analyzing the presence of inflammatory cells, mucus, bacterial flora, and epithelial cells. Only samples with a score of 4 or 5 were considered suitable for bacterial cultivation [44,45,46]. The PCR kits were selected based on the Gram stains: Gram-negative bacteria were tested using the Carba-R kit, and Gram-positive bacteria were tested using the MRSA/SA SSTI and van A/vanB kits, while mixed samples containing both Gram-negative and Gram-positive bacteria were tested using all three kits. The PCR kits were randomly assigned to the control group.

#### 4.2.2. Bacterial Cultures

Following the analysis of the Gram stain slides, a 10 μL sample from the ETA was inoculated onto corresponding culture media for bacterial isolation. The following media were used: sheep blood agar, chocolate agar, MacConkey agar, Chapman agar, and Sabourad dextrose agar (bioMerieux, Marcy-l’Étoile, France). The plates were incubated at 37 °C for 24 h. The bacterial growth was assessed according to the number of colony-forming units (CFU)/mL. A threshold of 10^5^ CFU/mL was set for positive samples.

A VITEK^®^ 2 (bioMerieux, Marcy-l’Étoile, France) antibiotic susceptibility testing system was used. A well-isolated bacterial colony was selected from the culture plate and emulsified in a sterile saline tube. The bacterial suspension was transferred onto specific cassettes following the manufacturer’s instructions, and automated susceptibility testing was performed. Antimicrobial testing was interpreted according to the Clinical and Laboratory Standards Institute (CLSI) [47].

#### 4.2.3. Identification of Bacteria or AMR Genes with GeneXpert

We used the Cepheid Xpert real-time PCR assay (Cepheid, Sunnyvale, CA, USA). This device is a POC multiplex PCR, which is easy to use and can deliver results in approximately one hour [25]. The current study used three assays: Xpert^®^ MRSA/SA SSTI, Xpert^®^ vanA/vanB, and Xpert^®^ Carba-R.

For the Xpert assay, according to the study protocol presented in Figure 1, we absorbed fluid from the same sample, which was used in the classic bacteriological identification process, into a device-specific cotton swab—ESwab (Copan, Brescia, Italy)—which was introduced in the extraction buffer, centrifugated according to the manufacturer’s instructions, and then transferred into the cartridge and placed into the GeneXpert device.

The Xpert^®^ MRSA/SA SSTI kit is a qualitative diagnostic test that detects the presence of methicillin-susceptible (MSSA) and methicillin-resistant *Staphylococcus aureus* (MRSA), validated and accepted for samples from skin and soft tissue. This assay is based on the simultaneous amplification and detection of three gene targets: the staphylococcal protein A (spa) gene, typical for *Staphylococcus aureus*; the gene for methicillin resistance (mecA) and the staphylococcal cassette chromosome (SCCmec), which contains the mecA gene inserted in the chromosomal *attB* situs; and the orfX from the *S. aureus* genome. The test is positive for MRSA only if all three targets are identified [48,49,50,51,52]. Targeting both the insertion place attB and the mecA gene permits the assay to identify variants with the deletion of the mecA gene, reducing the false-positive results, which can occur in the PCR tests that use only the SCCmec gene [49,53].

Xpert^®^ vanA/vanB qualitatively detects vancomycin-specific resistant genes from rectal and perianal swabs. It is mainly used to detect patients colonized with vancomycin-resistant bacteria, like *Enterococcus*. The assay has a high sensitivity for the detection of both vanA and vanB subtypes of genes typical for vancomycin-resistant *Enterococcus* (VRE) at very low bacterial loads (10–100 colony forming units/mL) [54,55].

Xpert^®^ Carba-R detects and differentiates five genes: bla_KPC_, bla_VIM_, bla_OXA-48_, bla_IMP-1_, and bla_NDM,_ which are associated with carbapenem resistance, also called metallo-beta-lactamases, in Gram-negative bacteria, mostly belonging to the *Enterobacterales* family. Also, this assay is only approved for usage from rectal swab specimens [56].

### 4.3. Statistical Analysis

The statistical analysis was designed and verified by a certified biomedical statistician. SPSS 28.0 (SPSSInc., Chicago, IL, USA) and MedCalc V22.007 (MedCalc Software Ltd., Ostend, Belgium) were used for the analysis. The sample size was calculated using an estimated 50% prevalence for a confidence interval of 95%. The statistical significance threshold was set as 0.05. Normally distributed variables were reported as mean ± standard deviation and compared using the Student’s *t*-test; variables with a non-normal distribution were reported as median (interquartile range) and compared with the Mann–Whitney U test. The diagnostic accuracy was reported in percentage and 95% confidence interval, using culture results as a reference standard.

### 4.4. Ethical Considerations

The Cluj-Napoca Emergency County Hospital’s Ethics Committee and the University of Medicine and Pharmacy ”Iuliu Hațieganu” approved the current research protocol. The protocol design complied with the ethical guidelines of the modified 1975 Declaration of Helsinki. Informed written consent was obtained from all subjects involved in the study on admission. Patient-related data were managed according to the European Union General Data Protection Regulation (GDPR).

## 5. Conclusions

Based on the current pilot study, rapid point-of-care PCR testing for drug-resistant mutations is an effective tool in facilitating antimicrobial stewardship, providing significantly quicker results with better sensitivity and specificity than the culture results.

## Figures and Tables

**Figure 1 ijms-24-13393-f001:**
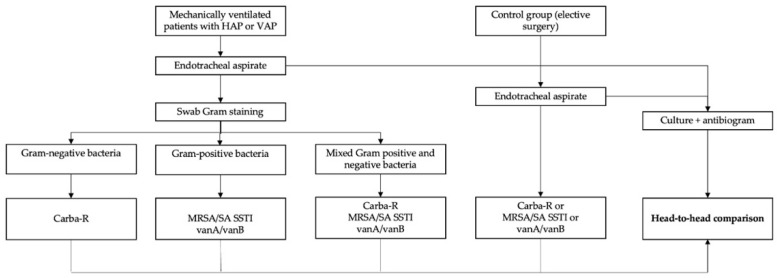
Study protocol algorithm. HAP—hospital-acquired pneumonia and VAP—ventilator-acquired pneumonia.

**Table 1 ijms-24-13393-t001:** The baseline characteristics of the study population.

	VAP/HAP Group (n = 29)	Control Group (n = 28)	*p* Value
Gender—male (n, %)	24 (82.75)	15 (53.57)	0.018
Age (years)	61.55 ± 18.42	59.07 ± 16.05	0.566
Length of ICU stay (days)	28 (14–48.5)	N/A	
Length of mechanical ventilation (hours)	336 (168–800)	N/A	
Rationale for ICU admission Aspiration pneumonia (n, %) Septic shock (n, %) Trauma (n, %) Other (n, %)	14 (48.27) 10 (34.48) 6 (20.68) 14 (48.27)	N/A	
SOFA	9.28 ± 3.99	N/A	
APACHE II	19.21 ± 7.52	N/A	
SAPS II	46.59 ± 14.31	N/A	
CPIS	6.45 ± 1.20	N/A	
White blood cell count (/mm^3^)	13,270 (10,660–18,910)	7700 (6210–11,230)	<0.001
Neutrophil count (/mm^3^)	12,700 (8650–22,350)	6700 (3560–8560)	<0.001
C-reactive protein (mg/dL)	19.49 ± 12.46	4.26 (1.23–9.39)	0.008
Presepsin (ng/L)	899 (218–1925)	N/A	
Procalcitonin (ng/mL)	1.31 (0.12–4.68)	0.46 (0.07–7.31)	0.006
Fibrinogen (mg/dL)	525 (381–650)	N/A	

Normally distributed variables are reported as mean ± standard deviation; variables with a non-normal distribution are reported as median (interquartile range); ICU—intensive care unit; SOFA—Sequential Organ Failure Assessment score; APACHE II—Acute Physiologic Assessment and Chronic Health Evaluation II scoring system; SAPS II—Simplified Acute Physiology Score; CPIS—Clinical Pulmonary Infection Score; N/A—not available.

**Table 2 ijms-24-13393-t002:** The bacterial infection profiles of the patients in the HAP/VAP group.

Primary Isolate	n = 29 (100%)
*Acinetobacter baumanii*	
MDR	6 (20.68%)
XDR	10 (34.48%)
*Klebsiella pneumoniae*	
MDR	2 (6.89%)
XDR (including CRE+)	2 (6.89%)
Methicillin-resistant *Staphylococcus aureus*	
MDR	2 (6.89%)
XDR (MSLb +)	1 (3.44%)
*Pseudomonas aeruginosa*	
MDR	4 (13.79%)
*Providencia stuartii*	
MDR	1 (3.44%)
XDR	1 (3.44%)
Secondary isolate	N = 5 (17.24%)
*Klebsiella pneumoniae*	
XDR (including CRE+)	4 (13.79%)
*Proteus* spp.	
MDR	1 (3.44%)

MDR—multidrug-resistant; XDR—extensively drug-resistant; CRE—Carbapenemase-producing *Enterobacteriaceae;* MSLb—macrolides, lincosamides, and streptogramins B resistance.

**Table 3 ijms-24-13393-t003:** Diagnostic test evaluation.

	All PCR Tests (n = 81)	Carbapenem-Resistance Kit (n = 47)	MRSA—Kit (n = 18)
Sensitivity	92.31% (79.13–98.38)	90.32% (74.25–97.96)	100% (54.07–100)
Specificity	97.67% (87.71–99.94)	95.65% (78.05–99.89)	91.67 (61.52–99.79)
Positive likelihood ratio	39.69 (5.71–275.99)	20.77 (3.04–141.75)	12 (1.84–78.37)
Negative likelihood ratio	0.08 (0.03–0.23)	0.10 (0.03–0.30)	0
Positive predictive value	97.30% (85.84—99.93)	96.55% (82.24–99.91)	85.71 (42.13–99.64)
Negative predictive value	93.33% (81.73–98.60)	88.00% (68.78–97.45)	100% (71.51–100)
Accuracy	95.12% (87.98–98.66)	92.59% (82.11–97.94)	94.44% (72.71–99.86%)

Results are expressed as estimate and 95% confidence interval.

## Data Availability

Data are contained within the article.

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
