# Peer review of "Rapid Point-of-Care PCR Testing of Drug-Resistant Strains on Endotracheal Aspirate Samples: A Repurposed Effective Tool in the Stepwise Approach of Healthcare-Acquired Pneumonia—A Pilot Study"

_ijms, 2023, doi:10.3390/ijms241713393_

Round 1
Reviewer 1 Report
The authors described a fast method to detect resistance-carrying bacteria during the recruitment of HCAP patients. The work overall may be fine but needs to be better described in the materials and methods section and in particular the whole bacterial growth and control phase to which little description has been devoted.
Author Response
Dear reviewer,
Thank you for taking the time to evaluate our manuscript. Indeed, we acknowledge that the Material and Methods section was overly simplified, as we mainly focused on describing the PCR technique. We have made substantial additions to the section in the revised version of the manuscript to provide a more in-depth description of the study protocol.
All the modifications are highlighted in red in the manuscript. For convenience and trackability, we decided against the use of the track-changes feature, as we believe that it actually makes the changes more difficult to follow amongst minor edits.
Hope that our response has met your expectations,
Kind regards,
The authors
Reviewer 2 Report
Rapid point-of-care PCR testing is a good topic as a test method that has recently attracted much attention. In particular, in the case of ICU-acquired pneumonia, it is a very useful study because rapid antibiotic resistance analysis must be performed. It is also meaningful to directly verify it with a sample of the lower respiratory tract.
As the author also mentioned, it is difficult to conclude with an analysis with too little sample amount. However, it is necessary to mention the advantages of field inspections and suggest that they need to be expanded to more samples in the future.
Author Response
Dear reviewer,
First of all, we would like to thank you for your appreciative comments. We have expanded the final paragraph of the Discussion section, incorporating your suggestions.
All the modifications are highlighted in red in the manuscript. For convenience and trackability, we decided against the use of the track-changes feature, as we believe that it actually makes the changes more difficult to follow amongst minor edits.
Hoping that our response has met your expectations,
Kind regards,
The authors
Round 2
Reviewer 1 Report
The authors have solved all my request.